# The Continuous Intention to Use E-Learning, from Two Different Perspectives

**Rana Saeed Al-Maroof [1], Khadija Alhumaid [2] and Said Salloum [3,***

1   English Language & Linguistics, Al-Buraimi University College, Al-Buraimi 512, Oman; rana@buc.edu.om
2   College of Education, Zayed University, Abu Dhabi, UAE; Khadija.Alhumaid@zu.ac.ae
3   Research Institute of Sciences and Engineering (RISE), University of Sharjah, Sharjah, UAE
*   Correspondence: ssalloum@sharjah.ac.ae

**Abstract:** During the recent vast growth of digitalization, e-learning methods have become the most influential phenomenon at higher educational institutions. E-learning adoption has proved able to shift educational circumstances from the traditional face-to-face teaching environment to a flexible and sharable type of education. An online survey was conducted, consisting of 30 teachers and 342 students in one of the universities in the United Arab Emirates. The results show that teachers' and students' perceived technology self-efficacy (TSE), ease of use (PEOU), and usefulness (PU) are the main factors directly affecting the continuous intention to use technology. Instructors' technological pedagogical content knowledge (TPACK) and perceived organizational support (POS) positively affect the intention to use the technology, whereas students' controlled motivation (CTRLM) has a greater influence on their intention to use the technology, due to the type of intrinsic and extrinsic motivation that they have and which they can develop throughout the process of learning. The findings support the given hypotheses. In addition, they provide empirical evidence of a relationship between perceived organizational support and perceived pedagogical content knowledge. In fact, they are considered the key factors that support the use of technology continuously.

**Keywords:** e-learning platform; PACK; perceived usefulness; perceived ease of use; perceived organizational support and technology self-efficacy

## 1. Introduction

Teachers and students may perceive the importance of e-learning differently. Teachers usually focus on the importance of training and support that may enhance the effective use of e-learning platforms, whereas the perceived usefulness and ease of use are influential factors from the students' perspective. The differences in their perspectives stem from the fact that their roles are different. Students usually receive the product through the e-learning system and can get all the different advantages that the system may offer; thus, they act as the consumers of the product [1]. On the other hand, teachers are the providers of the educational product, as they provide learners with the content and synthesize the given information in simple and concise language [1,2].

Past studies have focused on the importance of e-learning and its implementation all over the world; some have focused on the continuous use of e-learning [3–19] and some on the effect of either teachers' or students' attitudes towards e-learning [20–26]. In other words, no studies have put forward the implementation of two models that focus on how the perceived interactivity of education technology influences teachers' and students' perceptions and urges them to continue using the technology. This study assumes that some factors affecting teachers' intentions to use e-learning platforms continually are different from those affecting students' intentions to use the same platforms continuously. Therefore, this study proposes two different models that tackle both teachers' and students' continuous intentions to use technology. The two models will focus on a certain predictive power that has a more direct relationship with the teachers' and students' perceptions regarding

the continued use of the technology. For instance, one of the factors that contributes to the teachers' continuous use of the e-learning platform is the support they get from the university to enhance the use of the e-learning environment. On the other hand, one of the crucial factors for the students' perception is the controlled motivation that embraces certain intrinsic and extrinsic factors. It is worth mentioning that all previous studies have focused on students' perspectives on any technology-based technique. The acceptance of technology from a student's perspective has been dealt with in many papers, such as [27–29]. The fact that this study focuses on the effect of the same e-learning tool from a teacher's perspective separately is what sets this paper apart from other previous studies.

The objective of this study was to propose a theoretical framework that could be validated later through a proposed model that predicates the continuous intention to use e-learning among students at public universities in Dubai. There are numerous examples of literature related to technology acceptance [25,30–34] and continuous intention [35–38] that have been reviewed to identify the most common factors affecting the continuous intention to use the e-learning platform. The main concentration has been on theories that have been proven to have great predictive power in understanding users' perceptions and on theories that help to explain the importance of continuous use from two different perspectives. Hence, the Technological Pedagogical Content Knowledge (TPACK) was initiated by [39], Technology Acceptance Model (TAM) acceptance theory by [40], Social Cognitive Theory by [41], Perceived Organizational Support (POS) by [42], and Motivational Theory (MT) by [43,44]. The main factors that have been derived from these theories are perceived use and perceived usefulness, computer self-efficacy, controlled motivation, and so forth. The table below (Table 1) summarizes the studies that have tackled the continuous use of e-learning platforms.

**Table 1.** Most relevant studies of e-learning platforms in different sectors.

| Authors/ Reference | Target Population | Objective/Goal | Models Adopted |
|---|---|---|---|
| [45] | Students | To explain the f-variables that affect continued use of m-learning. | TAM, Theory of Planned Behavior (TPB), and Expectation Confirmation Model (ECM). |
| [46] | Students | To examine students' continuous use of blended learning, with reference to behavioral attitudes, motivations, and barriers. | TAM, TPB and self-determination theory (SDT). |
| [47] | Students | To make a connection between learners' adoption and satisfaction with LMS in blended learning in relation to certain learners' personal characteristics in terms of continuous use of the e-learning environment. | TAM and satisfaction factor (SAT). |
| [48] | Instructors | To examine the influential factors which may contribute to instructors' satisfaction with LMS use in a blended learning atmosphere. | LMS, system and instructors' characteristics that are derived from well-established factors. |
| [49] | Students | To investigate students' behavior of continuance intentions to use the double reinforcement interactive e-portfolio learning system. | TAM and IS continuance post-acceptance model (IS-TAM). |

**Table 1.** *Cont.*

| Authors/ Reference | Target Population | Objective/Goal | Models Adopted |
|---|---|---|---|
| [50] | Learners | To investigate the basic determinants behind the continuous intention to use e-learning. | TAM and Negative Critical Incident (NCI). |
| [51] | People chosen randomly through a high-traffic website | To investigate the motivational factors that affect the synthesized model that is composed of a combination of TAM, ECM, COGM and SDM. | TAM, ECM and cognitive model (COGM). |
| [52] | Technology users | To investigate and predict the main reason behind users' intentions to continue using e-learning. | ECM, TAM, and, TPB. |

As seen in the previous table showing the studies in the existing literature, much research has been conducted focusing on students and/or teachers within one proposed model. Nevertheless, searching for the predictive power behind both teachers' and students' intentions by proposing different variables is still neglected. To our knowledge, no research has examined the continued-use intention (CU) of teachers and instructors using e-learning platforms in higher education. Without knowledge of teachers' and students' CU, it is impossible to enhance e-learning in the Gulf area or to support its programs, systems, or administrative policies in terms of helping to sustain the e-learning platform.

## 2. Theoretical Framework and Hypotheses

The proposed framework has certain factors that can make the intention to use e-learning more measurable from two different perspectives. TPACK and POS are crucial elements that usually guide the teaching and learning environments from the teachers' perspectives. On the other hand, controlled motivation (CTRLM) is a factor that combines students' intrinsic and extrinsic motivations. Nevertheless, certain factors are equally important to both teachers and students, such as technology self-efficacy (TSE), perceived usefulness (PU), and perceived ease of use (PEOU).

### 2.1. Technological Pedagogical Content Knowledge (TPACK)

Teachers' knowledge cannot be tackled easily, as it is a complex concept that has many embedded elements [39]. The most important element is pedagogical content knowledge (PCK), which has been the domain of study for many researchers and practitioners. Its importance stems from the fact that it comprises both the content and the pedagogy that can explain how a particular topic is organized and how it is represented to the learners [53].

Since its emergence, TPACK has become a must since all teachers want to have a full understanding of the relationship between pedagogy and technology. TPACK refers to the type of technological pedagogical knowledge that teachers need to organize and present the intended teaching material effectively [39]. TPACK is one of the influential factors affecting teachers' perspectives. It refers to technological pedagogical content knowledge which includes: TCK (technological content knowledge), TPK (technological pedagogical knowledge), and PCK (pedagogical content knowledge) [54]. Self-assessment surveys and performance-based assessments are the basic instruments for evaluating TPACK [55,56].

The framework that comprises TPACK can be explained as combing different elements, such as content knowledge (CK), that highlight the teacher's knowledge of the subject matter. It includes knowledge of different types, such as knowledge of the theory, discipline, psychological aspects, historical aspects, and so forth [57]. The other two closely related elements are the pedagogy knowledge (PK), which is closely related to the teacher's

knowledge concerning methodology, process, and practice, and the pedagogical content knowledge, which is concerned with how teachers interpret and tailor the teaching material to suit certain pedagogical aims and purposes. This implies that the difference between the former element and the latter is the fact that the latter is related to methodology, assessment, and teaching style knowledge which can be used differently based on students' prior knowledge. Other elements are related to technology, as it comprises technology knowledge, technological content knowledge, and technological pedagogical knowledge. They are related to the ability of users to use technology to accomplish different tasks. The technological content knowledge is related to how technology can affect teaching material and vice versa. The final element is technological pedagogical knowledge, which has to do with the constraints that technology may impose on teaching material. This stems from the fact that certain technology is not developed for educational purposes and should be accustomed to suit educational purposes [57,58]. Accordingly, the following hypothesis can be formed:

**Hypothesis 1 (H1).** *TPACK will positively affect teachers' CU in the e-learning environment.*

### 2.2. Technology Self-Efficacy (TSE)

Self-efficacy is an effective factor that can reflect how students' own belief in their abilities to use technology affects their acceptance of the learning environment. Therefore, self-efficacy and learning are two factors that can, interactively and dynamically, affect each other [59]. Self-efficacy in the e-learning environment is considered an intrinsic motivator as far as continuous intention is concerned. It usually refers to the degree of confidence that users have in making use of technology [60]. Technology self-efficacy is usually identified as the ability to use technology without facing any crucial problems. It embraces two subdivisions: the estimation of result (users' estimations about their own input) and estimation efficacy (users' estimations in achieving the final result) [61–63]. Within the environment of e-learning, self-efficacy is highly connected to users' own beliefs regarding technology. Some believe that using technology is tremendously easy and achievable, while others may share a contradictory belief, as they may face problems in their ability to learn the appropriate way of using technology [64]. This simply implies that whenever users have a high level of technological self-belief, they may perceive the whole system properly; hence, they will be able to continue using the technology in a positive way. Accordingly, the following hypothesis may be formulated:

**Hypothesis 2 (H2).** *Technology Self-efficacy will positively affect teachers' and students' CU in the e-learning environment.*

### 2.3. Technology Acceptance Model (TAM)

A review of recent studies has shown that certain variables are crucial to understanding the reasons behind the continuous intention to use e-learning. Regarding Davis's TAM [65], it has been proven that PU and PEOU are the most influential factors in users' continuous-use intentions. Interestingly, PU is more effective than PEOU when one wants to deal with the use of technology [60]. This study focuses only on two constructs within the TAM theory, which have proven effective in investigating the continuous use of technology; these are perceived usefulness and ease of use. [65] adopted the view that the perceived usefulness (PU) and the perceived ease of use (PEOU) of technology form the baseline for examining individuals' usage intentions. PU is defined as the degree to which a person believes that using a technological system supports user performance, whereas PEOU tends to refer to the degree to which a person believes that use will be free of effort. Due to the fact that PEOU has proved to be of great significance only during the early-acceptance stage of technology use [54,65], PEOU may not directly affect teachers' and students' CU in the e-learning environment. Hence, we hypothesize the following:

**Hypothesis 3 (H3).** *The level of PU will positively affect teachers' and students' CU.*

**Hypothesis 4 (H4).** *The level of PEOU will not affect teachers' and students' CU.*

*2.4. Perceived Organizational Support (POS)*

Organizational support theory has a close relation with how users perceive organizational feedback regarding the use of technology. Users' perceptions may vary in accordance with an organization's rewards, fairness, and supervisor support [42]. To put it differently, when users had a positive attitude towards their organization, they were more willing to pursue their intention to use its technology and vice versa. The organization has a crucial role in enhancing the use of technology by motivating users internally at the organizational level [44,66]. Related literature is guided by the fact that organizational support for use of technology has a high impact on teachers' and students' CU to adhere to computer technology, particularly in technical support [67–69]. In this respect, teachers and students may have different subjective perspectives of the role of educational institutions (colleges and universities) in creating a motivational atmosphere regarding the continuous use of technology. Hence, it is hypothesized that if teachers and students had a positive perception of organizational support (POS), they would support the continuous usage of the technology. Thus, the following hypothesis may be formed:

**Hypothesis 5 (H5).** *The level of POS support will positively affect teachers' CU.*

*2.5. Controlled Motivation (CTRLM)*

According to [43], students' intrinsic and extrinsic motivations can be dealt with in terms of a hieratical model in which three factors play an influential role: contextual, situational, and global factors. One of the motivations that have a massive effect on user perception is called Controlled Motivation (CTRLM), which refers to a source of negative perception that is illustrated by the pressure that students may be under, both internally and externally. This type of pressure may lead to maladaptive outcomes, which are, in turn, illustrated by a combination of negative effects, perceived incompetence, and dissatisfaction [44]. A more updated view regarding controlled motivation is given by [44], who proposed two different types within CTRLM: introjected and external regulation. Introjected regulation has a close relationship with an individual's behavioral engagement, such as obligation, avoidance of guilt, ego-enhancement, and internal rewards. On the other hand, external regulation affects behavior engagement, including compliance, external rewards, and avoidance of punishments.

**Hypothesis 6 (H6).** *Controlled Motivation negatively affects students' continuous intention to use e-learning platforms.*

*2.6. The Proposed Research Models*

Based on the review of previous studies, it has been noticed that most of the studies on continuous intention have focused on one model that has factors that may be crucial to teachers but not students [60,70]. Therefore, this study attempted to build two models that can meet both teachers' and students' continuous intentions to meet e-learning demands. The proposed research models rely on these hypotheses, as illustrated in Figures 1 and 2.

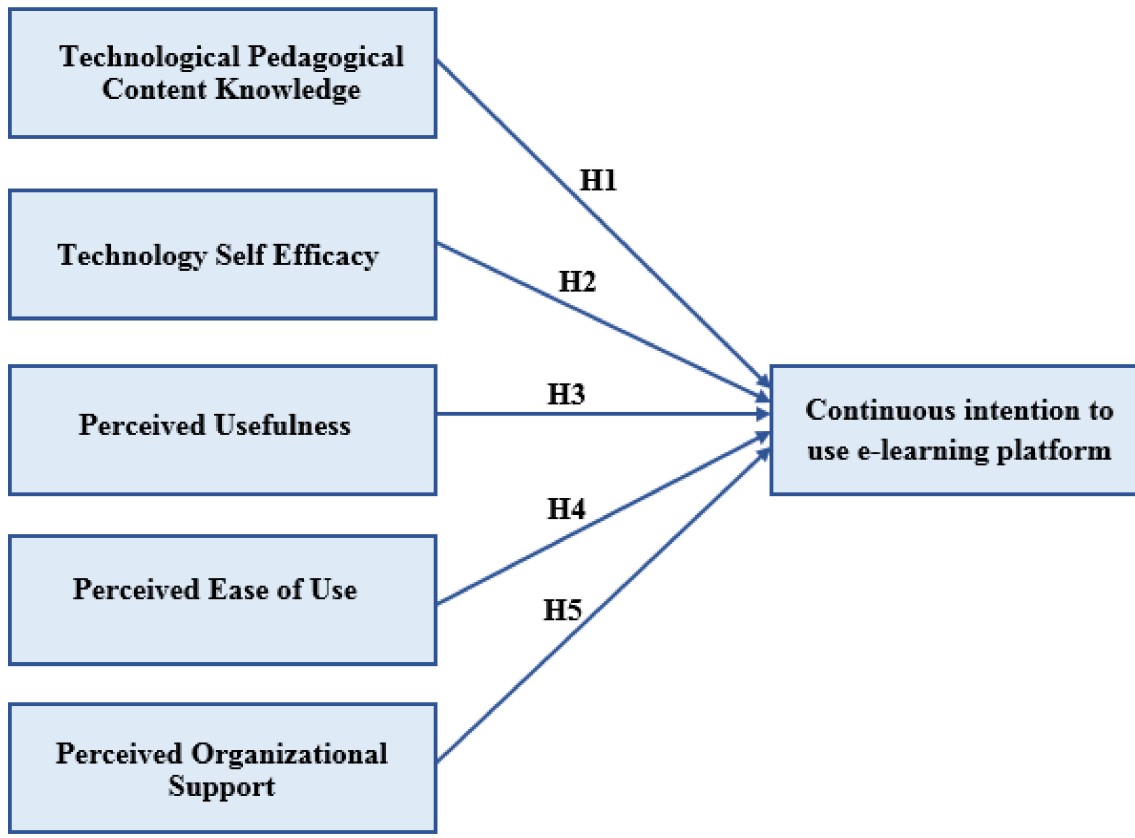

**Figure 1.** The e-learning technology model adopted for teachers. Note: H1–5 = Hypotheses 1–5.

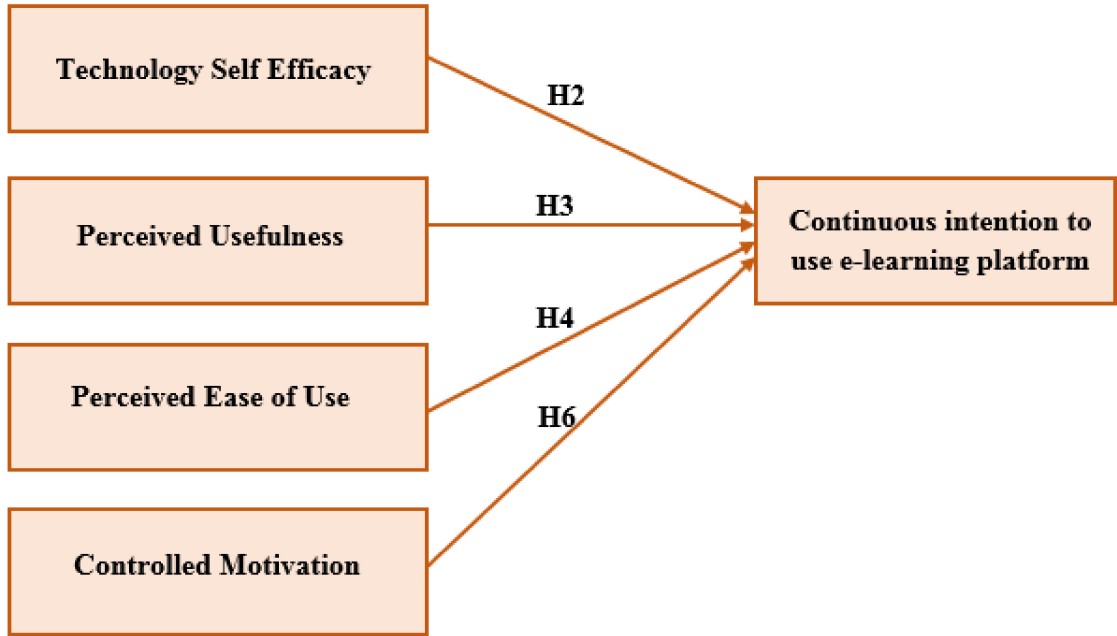

**Figure 2.** The e-learning technology model adopted for students.

## 3. Methodology

### 3.1. Participants

The participants (n = 372) were classified into two categories—teachers and students—in accordance with the two proposed models. This two-level selection was based on certain

influential factors. First, participants were identified as having sufficient experience in using e-learning platforms. Second, two different surveys were prepared and distributed to the two targeted groups of participants. The two surveys may have had similar and shared questions, but the teachers' survey may have had additional questions that the students' survey may have lacked. Responses were received from 372 participants. Therefore, the total number of teacher participants in this study was 30. These were college instructors with a nearly equal female-to-male gender ratio. On the other hand, the total number of student participants was 342. These were students at The British University in Dubai (BUiD).

### 3.2. Data Collection

During the winter semester of 2019/2020, the data was collected through online surveys from individuals studying at the British University in Dubai (BUiD) from 15 January to 20 February 2020. The aggregated response rate was 93%; 400 questionnaires were circulated, out of which 372 were answered by respondents. This means that 372 questionnaires were filled out correctly and found to be useful, while 28 were rejected because of missing values. The prospective sample size was 306 respondents with respect to a population of 1500. Thus, the sample size of 372 correct responses was suitable, according to [71], because—bearing in mind the required sample size—the sample size of 372 is a higher figure. Thus, this sample size could be reviewed using structural equation modeling [72] to verify the hypotheses. It must be noted that hypotheses were based on the current theories and were adjusted to the e-learning context. In order to assess the measurement model, the researchers used structural equation modeling (SEM) [72]. Further treatment was performed using a final path model.

### 3.3. Students' Personal Information/Demographic Data

The assessment of personal/demographic data is covered in Table 2. The percentage of males was 53%, while for females it was 47%. A total of 33% of students had ages ranging from 18 to 29 years, while 67% of respondents were aged over 29. In terms of academic background, 39% were students from the Faculty of Engineering and IT, 35% were from the Faculty of Education and 26% belonged to the Faculty of Business and Law. The majority of respondents came from sophisticated families and held university degrees; 49% of participants had bachelor's degrees, 42% had master's degrees, and 9% had a doctoral degree. When the respondents were ready to volunteer and were easily approachable, the purposive sampling approach was used as per [3]. This sample was created by students coming from different faculties, with different ages, enrolling in diverse programs at different levels. Moreover, with the aid of IBM SPSS Statistics ver. 23, the demographic data was evaluated. Table 2 depicts the complete demographic data of the respondents.

**Table 2.** Demographic data of the respondents.

| Criterion | Factor | Frequency | Percentage |
|---|---|---|---|
| Gender | Female | 175 | 47% |
| | Male | 197 | 53% |
| Age | Between 18 and 29 | 122 | 33% |
| | Between 30 and 39 | 98 | 26% |
| | Between 40 and 49 | 88 | 24% |
| | Between 50 and 59 | 64 | 17% |
| Faculties | Faculty of Engineering and IT | 145 | 39% |
| | Faculty of Education | 129 | 35% |
| | Faculty of Business and Law | 98 | 26% |
| Education qualification | Bachelor | 182 | 49% |
| | Master | 157 | 42% |
| | Doctorate | 33 | 9% |

### 3.4. Study Instrument

The survey instrument used to validate the hypothesis was determined in this research. The survey, consisting of 30 items, was used for the measurement of seven constructs in the questionnaire. Table 3 depicts the sources of the constructs. The questions from prior studies were modified in order to enhance the appropriateness of the research.

**Table 3.** Constructs and their sources.

| Constructs | Number of Items | Source |
|---|---|---|
| CU | 2 | [73–75] |
| CTRLM | 5 | [43] |
| TPACK | 4 | [39,76] |
| TSE | 7 | [41,77] |
| PEOU | 3 | [40] |
| PU | 4 | [40] |
| POS | 5 | [42] |

Note: TPACK = Technological pedagogical content knowledge; TSE = Technology self-efficacy; PEOU = Perceived ease of use; PU = Perceived usefulness; POS = Perceived organizational support; CTRLM = Controlled motivation; CU = Continuous intention to use e-learning platform.

### 3.5. Pilot Study for the Questionnaire

A pilot study was conducted to check the reliability of the questionnaire items. Approximately 40 students and teachers were chosen on a random basis from the given population to establish the pilot study. The sample size was set based on 10% of the aggregated sample size of this study (400 students and teachers) and thus adhered strictly to the research criteria. Cronbach's alpha test was utilized for the computation of internal reliability through IBM SPSS Statistics ver. 23, in order to judge the outcomes of the pilot study. Thus, the appropriate findings were shown for the measurement items. A value of 0.7 was taken to be an acceptable value for the reliability coefficient, considering the model for social science research [14]. Tables 4 and 5 show the Cronbach's alpha values for the seven measurement scales for teachers and students.

**Table 4.** Cronbach's alpha values for the pilot study (Cronbach's alpha $\geq$ 0.70) for teachers (Model A).

| Constructs | Cronbach's Alpha |
|---|---|
| CU | 0.756 |
| TPACK | 0.779 |
| TSE | 0.864 |
| PEOU | 0.889 |
| PU | 0.734 |
| POS | 0.852 |

Note: TPACK = Technological pedagogical content knowledge; TSE = Technology self-efficacy; PEOU = Perceived ease of use; PU = Perceived usefulness; POS = Perceived organizational support; CU = Continuous intention to use e-learning platform.

### 3.6. Survey Structure

The questionnaire survey given to students and teachers had two sections. Within the first part, personal data was given to gather information about students and teachers. The second section had a group of questions related to the main factors of the proposed models. The teachers' questionnaire had six sub-sections coinciding with the six factors proposed in the model. Similarly, the students' questionnaire had five sub-sections related to the five factors proposed in the model. With the help of the five-point Likert Scale, the 42

items were evaluated. The scales included the following: (1) strongly disagree, (2) disagree, (3) neutral, (4) agree, and (5) strongly agree.

**Table 5.** Cronbach's alpha values for the pilot study (Cronbach's alpha ≥ 0.70) for students (Model B).

| Constructs | Cronbach's Alpha |
|---|---|
| CU | 0.872 |
| CTRLM | 0.881 |
| TSE | 0.798 |
| PEOU | 0.736 |
| PU | 0.797 |

Note: TSE = Technology self-efficacy; PEOU = Perceived ease of use; PU = Perceived usefulness; CTRLM = Controlled motivation; CU = Continuous intention to use e-learning platform.

## 4. Findings and Discussion

### 4.1. Data Analysis

Along with the help of SmartPLS V.3.2.7 software, the partial least squares-structural equation modeling (PLS-SEM) was utilized to conduct the data analysis in this research [15]. The assessment approach had two steps of a structural model and a measurement model allowed to study the collected data [16]. There were various reasons for choosing PLS-SEM in the study. First, as the research is an extension of a current theory, PLS-SEM was considered the best option [17]. Second, the complex models within exploratory research can be effectively tackled with the help of PLS-SEM [18]. Third, PLS-SEM analyzes a complete model as a single unit, so there is no need to divide it [19]. Lastly, PLS-SEM provides concurrent analysis for measurement, as well as a structural model, leading to more accurate calculations [20].

### 4.2. Convergent Validity

In order to review the measurement model, it was suggested by [16] that construct reliability—including composite reliability (CR), Dijkstra–Henseler's rho (pA), and Cronbach's alpha (CA) and validity (including convergent and discriminant validity)—must be considered. Cronbach's alpha (CA) has values between 0.782 and 0.895, as Tables 6 and 7 show, in order to determine construct reliability. These statistics are higher than the threshold value of 0.7 [78]. According to Tables 6 and 7, the outcomes also show that the composite reliability (CR) has values from 0.796 to 0.882; these values are evidently bigger than the recommended value of 0.7 [79]. As an alternative, the construct reliability must be appraised by researchers by means of the Dijkstra–Henseler's rho (pA) reliability coefficient [80]. Like CA and CR, the reliability coefficient ρA must show 0.70 or higher in exploratory studies and values of more than 0.80 or 0.90 for further stages of research [78,81,82]. The reliability coefficient ρA of each measurement construct is above 0.70 according to Tables 6 and 7. According to these outcomes, the construct reliability is verified and all the constructs were considered to be accurate.

Convergent validity can be measured by testing the average variance extracted (AVE) as well as the factor loading [16]. Tables 6 and 7 suggest that all values of factor loadings exceeded the threshold value of 0.7. Moreover, Tables 6 and 7 show that the values obtained for the AVE were higher than the threshold value of 0.5, ranging from 0.509 to 0.718. Depending on the expected results, the convergent reliability can be obtained for all the constructs.

**Table 6.** Convergent validity results that ensure acceptable values (Factor loading, Cronbach's alpha, composite reliability (CR), Dijkstra–Henseler's rho (pA) ≥ 0.70 and average variance extracted (AVE) > 0.5) (Model A).

| Constructs | Items | Factor Loading | Cronbach's Alpha | CR | pA | AVE |
|---|---|---|---|---|---|---|
| Technology Self-Efficacy | TSE1 | 0.775 | 0.874 | 0.799 | 0.832 | 0.536 |
| | TSE2 | 0.736 | | | | |
| | TSE3 | 0.820 | | | | |
| | TSE4 | 0.901 | | | | |
| | TSE5 | 0.756 | | | | |
| | TSE6 | 0.723 | | | | |
| | TSE7 | 0.797 | | | | |
| Technological Pedagogical Content Knowledge | TPACK 1 | 0.711 | 0.829 | 0.882 | 0.791 | 0.552 |
| | TPACK 2 | 0.869 | | | | |
| | TPACK 3 | 0.909 | | | | |
| | TPACK 4 | 0.790 | | | | |
| Perceived Ease of Use | PEOU1 | 0.829 | 0.844 | 0.812 | 0.817 | 0.661 |
| | PEOU2 | 0.847 | | | | |
| | PEOU3 | 0.746 | | | | |
| Perceived Usefulness | PU1 | 0.734 | 0.816 | 0.828 | 0.825 | 0.623 |
| | PU2 | 0.766 | | | | |
| | PU3 | 0.889 | | | | |
| | PU4 | 0.850 | | | | |
| Perceived Organizational Support | POS1 | 0.729 | 0.863 | 0.814 | 0.883 | 0.718 |
| | POS2 | 0.848 | | | | |
| | POS3 | 0.758 | | | | |
| | POS4 | 0.819 | | | | |
| | POS5 | 0.878 | | | | |
| Continuous intention to use e-learning platform | CU1 | 0.796 | 0.815 | 0.876 | 0.898 | 0.673 |
| | CU2 | 0.801 | | | | |

**Table 7.** Convergent validity results that ensure acceptable values (Factor loading, Cronbach's alpha, composite reliability, Dijkstra–Henseler's rho ≥ 0.70 & AVE > 0.5) (Model B).

| Constructs | Items | Factor Loading | Cronbach's Alpha | CR | PA | AVE |
|---|---|---|---|---|---|---|
| Technology Self-Efficacy | TSE1 | 0.726 | 0.782 | 0.833 | 0.823 | 0.705 |
| | TSE2 | 0.826 | | | | |
| | TSE3 | 0.710 | | | | |
| | TSE4 | 0.868 | | | | |
| | TSE5 | 0.746 | | | | |
| | TSE6 | 0.733 | | | | |

**Table 7.** *Cont.*

| Constructs | Items | Factor Loading | Cronbach's Alpha | CR | PA | AVE |
|---|---|---|---|---|---|---|
| Perceived Ease of Use | PEOU1 | 0.763 | 0.895 | 0.800 | 0.836 | 0.559 |
| | PEOU2 | 0.890 | | | | |
| | PEOU3 | 0.849 | | | | |
| Perceived Usefulness | PU1 | 0.793 | 0.856 | 0.879 | 0.808 | 0.696 |
| | PU2 | 0.709 | | | | |
| | PU3 | 0.873 | | | | |
| | PU4 | 0.821 | | | | |
| Controlled Motivation | CTRL1 | 0.832 | 0.805 | 0.796 | 0.807 | 0.700 |
| | CTRL2 | 0.802 | | | | |
| | CTRL3 | 0.875 | | | | |
| | CTRL4 | 0.810 | | | | |
| | CTRL5 | 0.796 | | | | |
| Continuous intention to use e-learning platform | CU1 | 0.725 | 0.878 | 0.818 | 0.816 | 0.509 |
| | CU2 | 0.878 | | | | |

### 4.3. Discriminant Validity

The two criteria that were suggested should be measured to obtain the measurement of discriminant validity were the Fornell–Larcker measure and the Heterotrait–Monotrait ratio (HTMT) [16]. As per the findings of Tables 8 and 9, these needs have been verified by the Fornell–Larcker criterion as each AVE value, together with its square root, exceeds the value of the correlation of AVE with other constructs [83].

**Table 8.** Fornell–Larcker Scale (Model A).

| | TSE | TPACK | PEOU | PU | POS | CU |
|---|---|---|---|---|---|---|
| TSE | **0.876** | | | | | |
| TPACK | 0.165 | **0.845** | | | | |
| PEOU | 0.125 | 0.253 | **0.802** | | | |
| PU | 0.569 | 0.487 | 0.558 | **0.790** | | |
| POS | 0.187 | 0.202 | 0.291 | 0.115 | **0.787** | |
| CU | 0.369 | 0.198 | 0.378 | 0.383 | 0.178 | **0.803** |

Note: TPACK = Technological pedagogical content knowledge; TSE = Technology self-efficacy; PEOU = Perceived ease of use; PU = Perceived usefulness; POS = Perceived organizational support; CU = Continuous intention to use e-learning platform.

**Table 9.** Fornell–Larcker Scale (Model B).

| | TSE | PEOU | PU | CTRL | CU |
|---|---|---|---|---|---|
| TSE | **0.768** | | | | |
| PEOU | 0.368 | **0.801** | | | |
| PU | 0.267 | 0.229 | **0.887** | | |
| CTRL | 0.649 | 0.492 | 0.399 | **0.844** | |
| CU | 0.422 | 0.327 | 0.302 | 0.188 | **0.870** |

Note: TSE = Technology self-efficacy; PEOU = Perceived ease of use; PU = Perceived usefulness; CTRLM = Controlled motivation; CU = Continuous intention to use e-learning platform.

Tables 10 and 11 show the outcomes of the HTMT ratio, indicating that the threshold value of 0.85 is bigger than the values of other constructs [27], and hence confirming the HTMT ratio. These outcomes play a role in the evaluation of the discriminant validity. The outcomes of the analysis indicated a smooth and simple assessment of the measurement model in terms of the model's validity and reliability. To conclude, it can be said that the collected data was appropriate for additionally evaluating the structural model.

**Table 10.** Heterotrait–Monotrait Ratio (HTMT) (Model A).

|       | TSE   | TPACK | PEOU  | PU    | POS   | CU  |
|-------|-------|-------|-------|-------|-------|-----|
| TSE   |       |       |       |       |       |     |
| TPACK | 0.560 |       |       |       |       |     |
| PEOU  | 0.136 | 0.487 |       |       |       |     |
| PU    | 0.266 | 0.363 | 0.556 |       |       |     |
| POS   | 0.296 | 0.200 | 0.270 | 0.544 |       |     |
| CU    | 0.389 | 0.635 | 0.378 | 0.638 | 0.555 |     |

Note: TPACK = Technological pedagogical content knowledge; TSE = Technology self-efficacy; PEOU = Perceived ease of use; PU = Perceived usefulness; POS = Perceived organizational support; CU = Continuous intention to use e-learning platform.

**Table 11.** Heterotrait–Monotrait Ratio (HTMT) (Model B).

|      | TSE   | PEOU  | PU    | CTRL  | CU  |
|------|-------|-------|-------|-------|-----|
| TSE  |       |       |       |       |     |
| PEOU | 0.232 |       |       |       |     |
| PU   | 0.506 | 0.436 |       |       |     |
| CTRL | 0.392 | 0.457 | 0.503 |       |     |
| CU   | 0.697 | 0.609 | 0.210 | 0.264 |     |

Note TSE = Technology self-efficacy; PEOU = Perceived ease of use; PU = Perceived usefulness; CTRLM = Controlled motivation; CU = Continuous intention to use e-learning platform.

### 4.4. Model Fit

The subsequently mentioned fit measures are ensured by SmartPLS: the standard root mean square residual (SRMR), exact fit criteria, Euclidean distance (d_ULS), geodesic distance (d_G), Chi-square, Normed Fit Index (NFI), and RMS Theta show the model fit in PLS-SEM [84]. The difference between experimental correlations and the correlation matrix inferred from model [85] are indicated by SRMR, and values smaller than 0.08 are assumed to serve as good model-fit measures [86]. The NFI values that are higher than 0.90 point out a good model fit [87]. The NFI is a ratio of the Chi-square value of the proposed model to the null model (also known as the benchmark model) [88]. The NFI increases with larger parameters and therefore, the NPI is not suggested as a model-fit pointer [85]. The discrepancy between the empirical covariance matrix and the covariance matrix, inferred from the composite factor model, is indicated by the metrics of squared Euclidean distance (d_ULS) and the geodesic distance (d_G) [80,85]. The RMS Theta helps in the measurement of the degree of outer model residuals correlation and is appropriate for reflective models only [88]. The nearer the RMS Theta value is to zero, the more superior the PLS-SEM model, and their values of less than 0.12, are assumed to be a good fit, with anything other than this suggesting an absence of fit [89]. The saturated model evaluates the correlation between all constructs, as recommended by [29], while the approximate model takes all the effects and model structure into consideration. The RMS Theta value was 0.073 in Model A and 0.073 in Model B, as given in Tables 12 and 13, which gives an idea that the specific goodness-of-fit for the PLS-SEM model was big enough to prove global PLS model validity.

**Table 12.** Model fit indicators (Model A).

| | Complete Model | |
|---|---|---|
| | **Saturated Model** | **Estimated Mod** |
| SRMR | 0.031 | 0.041 |
| d_ULS | 0.786 | 3.216 |
| d_G | 0.565 | 0.535 |
| Chi-Square | 466.736 | 473.348 |
| NFI | 0.624 | 0.627 |
| RMS Theta | 0.073 | |

**Table 13.** Model fit indicators (Model B).

| | Complete Model | |
|---|---|---|
| | **Saturated Model** | **Estimated Mod** |
| SRMR | 0.012 | 0.031 |
| d_ULS | 0.605 | 2.317 |
| d_G | 0.516 | 0.506 |
| Chi-Square | 461.646 | 472.347 |
| NFI | 0.633 | 0.642 |
| RMS Theta | 0.061 | |

*4.5. Hypotheses Testing Using PLS-SEM*

The interdependence between different theoretical constructs related to the structural model was studied by using a combination of the structural equation model with maximum-likelihood estimation and SmartPLS [38,39]. This indicates the procedure of analysis of the proposed hypothesis. About 83% and 71% variance were found within the continuous intention to use the e-learning platform, as shown in Tables 14 and 15, which indicates a high predictive power of Models A and B [37]. For all the proposed hypotheses, outcomes of the PLS-SEM technique provided the beta (β) values, *t*-values, and *p*-values, which have been stated in Tables 16 and 17. It is evident that each and every hypothesis is supported by all the researchers. The empirical data backs hypotheses H1, H2, H3, H4, H5, and H6 on the basis of the analyzed data. The standardized path coefficients and path significances are demonstrated in Figures 3 and 4.

**Table 14.** $R^2$ of the endogenous latent variables (Model A).

| **Constructs** | $R^2$ | **Results** |
|---|---|---|
| Continuous intention to use e-learning platform | 0.832 | High |

**Table 15.** $R^2$ of the endogenous latent variables (Model B).

| **Constructs** | $R^2$ | **Results** |
|---|---|---|
| Continuous intention to use e-learning platform | 0.709 | High |

In Model A, technological pedagogical content knowledge (TPACK), technology self-efficacy (TSE), perceived ease of use (PEOU), perceived usefulness (PU), and perceived organizational support (POS) have significant effects on continuous intention to use the e-learning platform (CU) (($\beta = 0.336$, $p < 0.001$), ($\beta = 0.426$, $p < 0.05$), $\beta = 0.589$, $p < 0.05$), ($\beta = 0.625$, $p < 0.05$) and ($\beta = 0.553$, $p < 0.001$), respectively); hence, H1, H2, H3, H4, and H5 are supported.

In Model B, technology self-efficacy (TSE), perceived ease of use (PEOU), perceived usefulness (PU), and controlled motivation (CTRLM) have significant effects on continuous intention to use the e-learning platform (CU) (($\beta$ = 0.290, $p$ < 0.001), ($\beta$ = 0.357, $p$ < 0.05), $\beta$ = 0.465, $p$ < 0.05) and ($\beta$ = 0.243, $p$ < 0.05), respectively); hence, H2, H3, H4, and H6 are supported.

**Table 16.** Hypotheses testing of the research model (significant at ** $p$ < = 0.01, * $p$ < 0.05) (Model A).

| H | Relationship | Path | *t*-Value | *p*-Value | Direction | Decision |
|---|---|---|---|---|---|---|
| H1 | TPACK -> CU | 0.336 | 12.223 | 0.001 | Positive | Supported ** |
| H2 | TSE -> CU | 0.426 | 5.269 | 0.026 | Positive | Supported * |
| H3 | PU -> CU | 0.589 | 6.716 | 0.018 | Positive | Supported * |
| H4 | PEOU -> CU | 0.625 | 5.584 | 0.023 | Positive | Supported * |
| H5 | POS -> CU | 0.553 | 16.108 | 0.000 | Positive | Supported ** |

Note: TPACK = Technological pedagogical content knowledge; TSE = Technology self-efficacy; PEOU = Perceived ease of use; PU = Perceived usefulness; POS = Perceived organizational support; CU = Continuous intention to use e-learning platform.

**Table 17.** Hypotheses testing of the research model (significant at ** $p$ < = 0.01, * $p$ < 0.05) (Model B).

| H | Relationship | Path | *t*-Value | *p*-Value | Direction | Decision |
|---|---|---|---|---|---|---|
| H2 | TSE -> CU | 0.290 | 14.578 | 0.000 | Positive | Supported ** |
| H3 | PEOU -> CU | 0.357 | 3.116 | 0.043 | Positive | Supported * |
| H4 | PU -> CU | 0.465 | 2.646 | 0.035 | Positive | Supported * |
| H6 | CTRLM -> CU | 0.243 | 4.361 | 0.033 | Positive | Supported * |

Note: TSE = Technology self-efficacy; PEOU = Perceived ease of use; PU = Perceived usefulness; CTRLM = Controlled motivation; CU = Continuous intention to use e-learning platform.

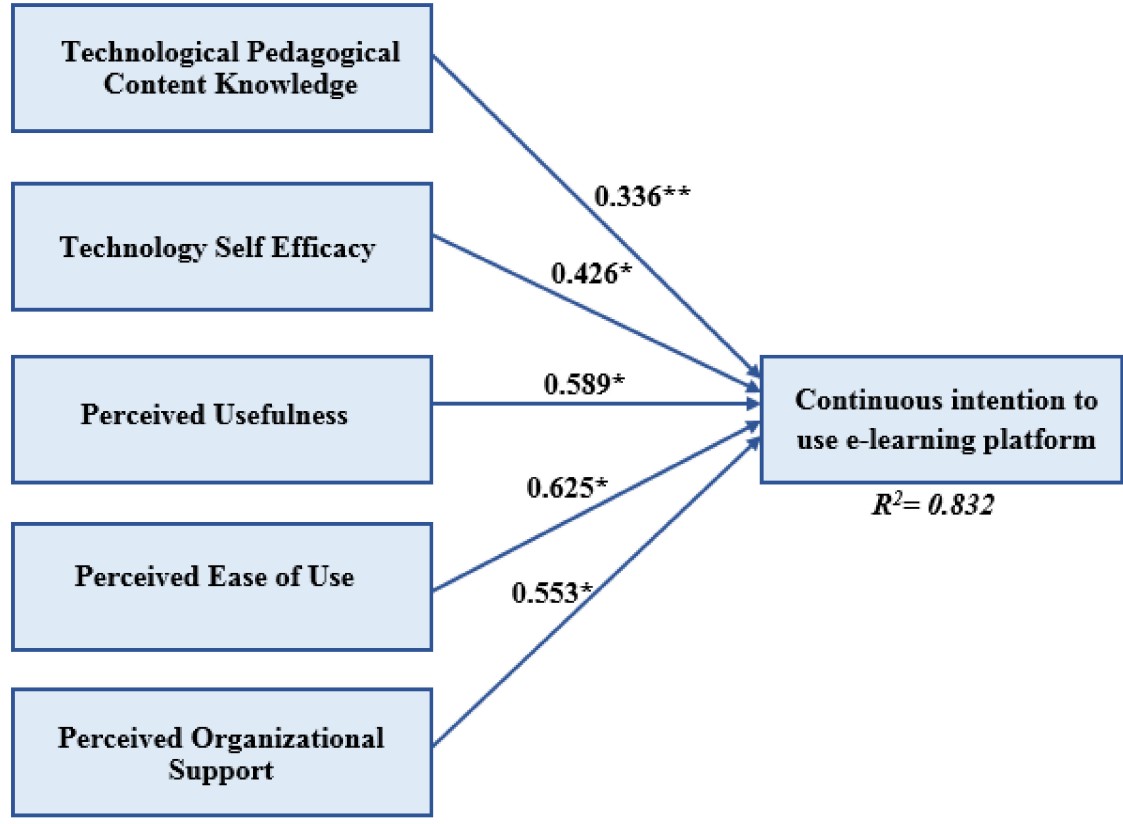

**Figure 3.** Path coefficient of the model (significant at ** $p$ <= 0.01, * $p$ < 0.05) (Model A).

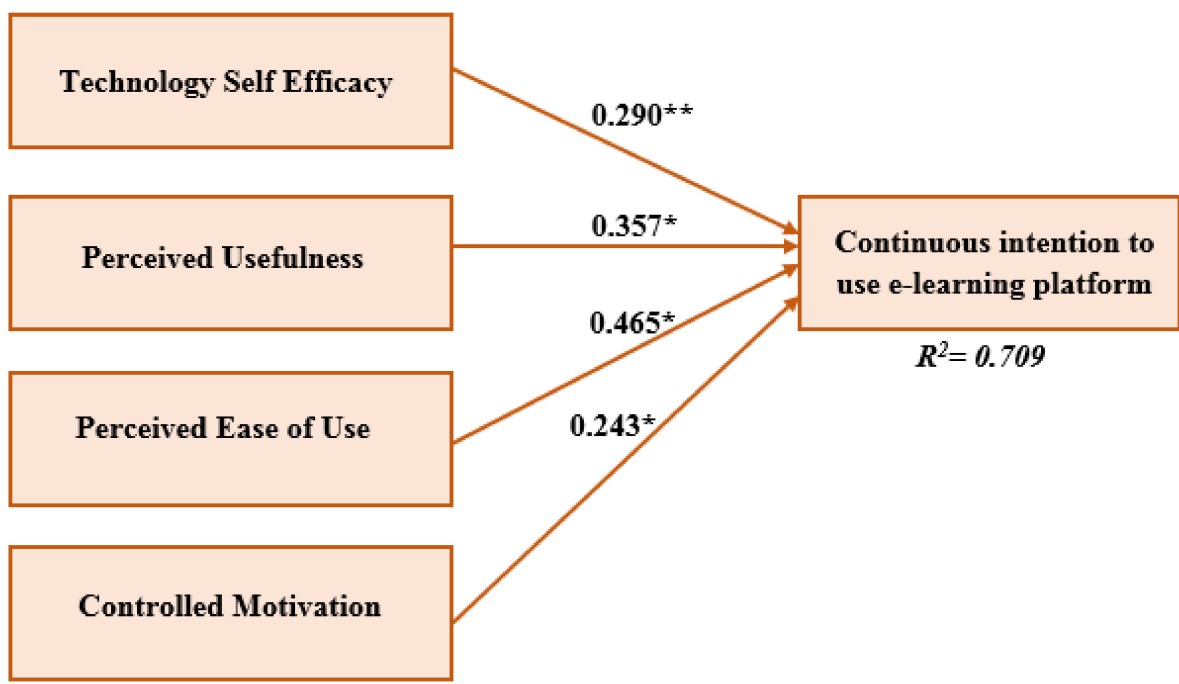

**Figure 4.** Path coefficient of the model (significant at ** $p < = 0.01$, * $p < 0.05$) (Model B).

### 5. Discussion and Conclusions

This study proposed two unique CU models that took into consideration factors that affect both instructors' and students' attitudes. The two models can be theoretically extended to enhance other technology-supported educational environments and instructional processes. The first research model of instructors' CU was proposed, taking into consideration certain social cognitive theory along with personal, behavioral, and environmental elements that are closely related to instructors' CU. In general, the results of SEM analysis has supported all the proposed hypothesis. From a practical perspective, this study has proven that POS is the most influential factor that affects instructors' CU of e-learning platforms.

Ref. [90–92] seem to agree with the current conclusion in stating that POS could motivate its staff members, leading to an upgrade of the organization. However, a study by [93] placed an emphasis on staff members' personalities and readiness to change. This implies that the lack of organizational support may have negative consequences. One of the studies by [60] has proposed that when instructors feel that there is no adequate organizational support, they are less likely to continuously use the technology, especially in an educational atmosphere where instructors are supposed to implement various in-class pedagogical changes to facilitate a better learning environment for the students.

In fact, POS is not the only factor that affects instructors' CU, but rather instructors' TPACK is another key factor that affects CU. Most of the previous research has proven that the organizational support may affect users' motivation to use the technology. A study by [94] put emphasis on the effect of PACK in facilitating the e-learning process by both teachers and adult learners. This seems to be in line with the results obtained from this study where TPACK affected, to great extent, the teachers' performance. It is assumed that whenever teachers' content, technological, and pedagogical knowledge is high, it implies that his or her ability to change the teaching material to suit the newly used technology will be more practical and effective. Obviously, teachers are more motivated when there is reliable technical support and IT staff that can facilitate the process of establishing new computer-supported knowledge [66,68].

Similarly, CTRLM has an effect on students' CU. CTRLM, along with technology self-efficacy (TSE), are the key factors that have a great impact on students' CU. The study has proven that CTRLM is connected deeply with the willingness to use the e-learning teaching platform. The higher the motivation is, the more effective results are obtained. Previous studies have tackled the effect of CTRLM on students' performance and have proven that there are both pedagogical and non-pedagogical elements that affect motivation [95–97]. These studies have indicated that technology development has placed a positive effect on motivation, as it urges students to get engaged in the new learning platform. This makes motivation very high and involvement in classes even higher, increasing the students' willingness to learn, and thus, using the technology continuously [98,99].

### 5.1. Practical Implications

According to the study outcomes, salient factors were noticed in determining users' acceptance of the e-learning system and technological pedagogical content knowledge, technology self-efficacy, perceived ease of use, perceived usefulness, perceived organizational support, and controlled motivation were significant. Hence, upon implementing a new e-learning system, faculty members must be informed about the system's features and its technical issues, as well as its usefulness, so that they feel self-assured and can gain insight into the system. To increase faculty use of the e-learning system, faculty members are of the view that universities should deliver workshops, extensive training, and awareness programs on the system's features, benefits, and usage [100]. In addition, a national survey was conducted, in which 57% of faculty members said that they could become more productive [100] if the use of e-learning technology was communicated thoroughly to them in their courses. Faculty members in this study also reported that, if they were aware of the positive impact of such technology on student learning, they would be inspired to learn and use the e-learning system. Hence, extended online help and periodic training programs for the e-learning system should be offered by the universities to ensure increased use of the system and to increase the faculty's self-efficacy. As a result, this overall phenomenon would help faculty members to obtain practical exposure, acquire better skills and become more proficient in using the e-learning system; as a result of this, their use of the system would be enhanced. In accordance with our findings, there was a weak influence on faculty attitudes toward the e-learning system because of facilitating conditions. Consequently, to ensure the smooth running of the e-learning system, attention must be given by universities to ensuring technical support and reliable network access. Moreover, online and face-to-face support and guidance should be provided by the universities for faculty members to ensure that members have positive attitudes toward the e-learning system and, consequently, that they become capable of extended use of the system [65,101].

### 5.2. Limitations and Further Research

This study has the following limitations. Only one university in the United Arab Emirates was considered for studying the impact of factors on the adoption of e-learning systems; this is the key limitation. The applicability and pertinence of this study would have been enhanced if more institutes and universities in the United Arab Emirates had been taken into consideration. With further analysis and insight into the e-learning system, the researchers could have better understood the factors that influence such a system. In addition, the participation of a limited number of students (372) is another limitation. According to [45], a survey questionnaire method was used for data collection. For appropriate and improved outcomes, an enhanced instrument is being sought and more institutes will be taken into account from other regions, such as the Arab Gulf, including Kuwait, Saudi Arabia, and Bahrain. Furthermore, invitations to join the study will be sent to more students, and researchers will conduct focus groups and interviews for suitable results. Moreover, we are seeking to implement the e-learning system in specific Arab universities that have contributed to the research.

**Author Contributions:** Conceptualization, R.S.A.-M., K.A. and S.S.; methodology, R.S.A.-M., K.A. and S.S.; software, S.S.; validation, S.S.; formal analysis, R.S.A.-M., K.A. and S.S.; investigation, R.S.A and K.A.; resources, K.A.; data curation, S.S.; writing—original draft preparation, R.S.A.-M.; writing—review and editing, K.A. and S.S.; visualization, R.S.A.-M., K.A. and S.S.; supervision, R.S.A.-M., K.A. and S.S.; project administration, R.S.A.-M., K.A. and S.S. All authors have read and agreed to the published version of the manuscript.

**Funding:** This research received no external funding.

**Institutional Review Board Statement:** Not applicable.

**Informed Consent Statement:** Informed consent was obtained from all subjects involved in the study.

**Conflicts of Interest:** The authors declare no conflict of interest.

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
