# Peer review of "The Continuous Intention to Use E-Learning, from Two Different Perspectives"

_education, doi:10.3390/educsci11010006_

Round 1

Reviewer 1 Report

The purpose of the study was to investigate the motivation behind tech adoption in elearning from the instructor and sutdent perspective. Work is similar to a theoretical analysis on adopting change in education, using a conventional survey and theory modeling approach using sem. The study is described in great detail and contributes to the literature, I support its publication but would like the authors address three questions I had about the study. 1) The study collapses 30 instructors with 340 students. Is 30 instructors sufficient in SEM for the factors converged related to instructors for model a? Given the complexity of the concepts presented, I would suggest a paper only investigating the student results to allow elaboration on the theories applied. I am open to changing my mind if there is evidence in the study that can be shown to support the reliability of the teacher's perception towards the model, 2) Although the alphas look great for scales with 3-7 likert items for each construct, what are the evidence for the items representing the respective constructs( I know you showed table 9, but how would I know TSE is TSE? Were the items extracted from validated instruments?), and how were those items developed? For example, I read through Mishra and Koehler's and Grandgenett's work on TPCK and cannot find the 4 items related to its framework. Could you present the items as written? 3) the study used a 42 item survey, but table 3 described 30 items. What were the remaining items? A subgroup analysis between faculty of the students or learner level would have been nice because the current assumption combines learners at different levels of education.

If these questions can be addressed I think this study should be published.

Author Response

Dear Prof.

The authors are really very grateful to the Journal Editor efforts in essentially discussing the paper reviews with the reviewers and identifying the further significant points for enhancements. The authors would also like to sincerely thank the reviewers for their valuable remarks and careful feedback which helped them to significantly enhance this work and its presentation. Regardless of the final outcome, the authors sincerely thank the editor and reviewers for supporting them. The productive and valuable remarks enable them to update many parts of the paper as shown by the responses to each comment. Besides, all the updated parts in the manuscript were highlighted in yellow color in order to be easily tracked by the editor and reviewers.

Best regards,

Reviewer 2 Report

The article is of interest but requires some changes to be made.

Abstract

Anticipate some of the results obtained, the main results.

Prepare the abstract in the format: Introduction, Method, Results and Discussion.

Introduction
The objectives and purposes of the study are well defined and contextualized

The Hypotheses are well formulated for both teachers and students

Method
Participants
30 teachers (reduced number of teachers in the sample). There are 342 students, with a larger sample size the external validity of the study would have been greater.

Instrument
They use only 1 questionnaire, only one instrument.
The pilot study used 40 subjects, for an instrument of 42 items (perhaps few) for the validation of the instrument.

The reliability of the scores, alpha coefficient, is located between 0.7 and 0.8, both for teachers and students, the values ​​are relevant, without being high.

The authors make mention of intrinsic motivation and extrinsic motivation, but no instrument is used for their assessment. They consider the intrinsic motivation and extrinsic motivation that they have and can develop throughout the learning process. Therefore, if they continue investigating in this line, they should use a motivational instrument to establish its relationship with the scores obtained.

Data analysis
A least squares-structural equation modeling (PLS-SEM) is used, being a successful procedure.
But they report the analysis of the data in a section other than the method, within the Method should be analyzed: Participants, Instruments, Procedure and Analysis of the data.

Boards
The tables have excessive interior lines, they do not respond to the rules of the articles, as they have too many interior squares. The authors must modify this aspect.

Discussion and Conclusions

There are conclusions but not a detailed discussion, in which a deep discussion with other articles of recognized prestige appears. The discussion should be increased with citations to other studies and comparison of the results obtained with the evidence from other studies.

Lines of improvement are established as well as practical implications and the article ends with a successful conclusion.
But it is necessary to carry out a greater discussion of the results obtained with some international studies, to use more citations in the discussion, comparing the evidence obtained with the results of other studies.

References

43 of the 88 references are from the last 5 years (50%) thematic news index is correct.

Author Response

(The authors gave the same response as above.)

Reviewer 3 Report

  1. This paper explored to models regarding e-learning platforms for teachers and students. This is an intriguing topic given the work and study from home circumstances. However, regarding the model, I have a few questions. The proposed model displayed a direct path from technological pedagogical content knowledge, technology self-efficacy, perceived usefulness, perceived ease of use, and perceived organizational support to continuous intention to use e-learning platform.  Similar constructs for students. So I'm wondering why these construct were not correlated in the model. For example, in the student model, self efficacy and motivation are correlated as examined in other researches.  
  2. The author suggested that the instrument used to collect students data were modified in order to enhance the appropriateness of the research. Therefor, it would be better to provide more information on what kind of modifications were made to adjust the survey. 
  3. In addition to that, the process to collect the data is not very clear. For example, is the questionnaire collected online via Qualtrics or other online survey platform?
  4.  

Author Response

(The authors gave the same response as above.)
